# Long Noncoding RNA, MicroRNA, Zn Transporter Zip14 (Slc39a14) and Inflammation in Mice

**DOI:** 10.3390/nu14235114

**Published:** 2022-12-01

**Authors:** Felix R. Jimenez-Rondan, Courtney H. Ruggiero, Robert J. Cousins

**Affiliations:** Center for Nutritional Sciences and Food Science and Human Nutrition Department, University of Florida, Gainesville, FL 32611, USA

**Keywords:** zinc, epigenetics, organoids, intestine, zinc transporter

## Abstract

Integration of non-coding RNAs and miRNAs with physiological processes in animals, including nutrient metabolism, is an important new focus. Twenty-three transporter proteins control cellular zinc homeostasis. The transporter Zip14 (Slc39a14) responds to proinflammatory stimuli. Using enterocyte-specific *Zip14* knockout mice and RNA-sequencing and quantitative polymerase chain reaction (qPCR), we conducted transcriptome profiling of proximal small intestine, where Zip14 is highly expressed, using RNA from whole intestine tissue, isolated intestinal epithelial cells (IECs) and intestinal organoids. *H19, U90926, Meg3, Bvht, Pvt1, Neat1* and *miR-7027* were among the most highly expressed genes. Enterocyte-specific deletion of *Zip14* demonstrated tissue specific expression, as such these changes were not observed with skeletal muscle. Chromatin immunoprecipitation (ChIP) assays of chromatin from isolated intestinal epithelial cells showed that enterocyte-specific *Zip14* deletion enhanced binding of proinflammatory transcription factors (TFs) signal transducer and activator of transcription 3 (STAT3) and nuclear factor kappa beta (NF-ĸβ) to promoters of *H19*, *Meg3* and *U90926*. We conclude enterocyte-specific ablation of *Zip14* restricts changes in those RNAs to the intestine. Binding of proinflammatory TFs, NF-ĸβ and STAT3 to the *H19*, *Meg3* and *U90926* promoters is consistent with a model where *Zip14* ablation, leads to increased TF occupancy, allowing epigenetic regulation of specific lncRNA genes.

## 1. Introduction

There is robust literature on long non-protein-coding RNA (lncRNA) and small non-coding RNA or microRNA (miRNA) associated with an array of biological processes. The abundance of current information on lncRNAs and miRNAs stems to a considerable degree on data sets from RNA-sequencing and other transcriptome screening platforms. These efforts illustrate the roles of non-coding RNAs associated with necessary cellular functions. In most cases, their functions influence regulatory pathways and gene expression necessary for normal cell function. Atypical or excessive abundance of lncRNAs and miRNAs appear to be associated with disease, either extensions of normal functions or abnormal departures to disrupt normal homeostatic mechanisms [1,2,3,4].

An important focus of non-coding RNAs is their association with necessary cellular functions, which have developmental and physiological outcomes in animals. Expectedly, this includes nutrient metabolism. Early examples of such roles are the influence of miR-122 in iron homeostasis [5] and miR-33 for cholesterol transport [6]. Zinc, an essential micronutrient, has documented roles in biology that are classified as structural, catalytic and regulatory [7]. Frequently they overlap, e.g., the zinc finger motifs of proteins maintain structure, but also influence gene expression. Similarly, most histone deacetylase enzymes (HDACs) are zinc-dependent and require zinc for catalytic activity, yet they function to influence gene expression [8,9]. Our interest in zinc metabolism and status assessment stimulated our initial research on identification of zinc-responsive miRNAs. Using serum episomes from human subjects that were fed a zinc-deficient diet under controlled conditions, we found that a select sub-set of miRNAs were influenced by zinc intake [10]. As detected by quantitative polymerase chain reaction (qPCR), this sub-set of miRNAs decreased with dietary zinc restriction and returned to normal abundance levels within days with acute zinc repletion. These experiments were never followed up, despite the fact that they respond to dietary zinc restriction in humans and there is a need for a biomarker of zinc status [7]. Other investigators have described studies associating miRNAs/lncRNAs with zinc, including zinc effects on prostate cancer cell lines [11], transporter Zip9 and radiation-induced skin damage [12], transporter Zip7 and gastric cancer [13], ZnT4 and acute cerebral ischemia [14], human deficiency and esophageal cancer [15], and Zip8 and osteoarthritis [16].

To broaden our understanding of the influence of cellular zinc trafficking in skeletal muscle, we conducted experiments using a murine global knockout for the metal transporter Zip14 (Slc39a14). Transcript profiling with arrays resulted in the relative expression of the lncRNA, *H19*, and a derivative miRNA, *miR-675*, prominently upregulated in the whole body *Zip14* knockout mice (WB-KO) [17]. Skeletal muscle from these genotypes shows muscle wasting, accompanied by decreased zinc uptake. The mice with the *Zip14* ablation had a proinflammatory phenotype. High relative expression of *H19/miR-675* was found in muscle of the WB-KO mice. Moreover, *Zip14* and *H19/miR-675* were induced in muscle with Lipopolysaccharide (LPS) treatment. Differential expression of specific transcripts was positively related to expression changes of specific lncRNAs/miRNAs, as a result of deletion of *Zip14* and a reduction in zinc transport activity.

Here we show comparative effects of whole body *Zip14* deletion on lncRNA/miRNA expression in skeletal muscle and proximal small intestine, as well as experiments with lncRNA/miRNA expression differences in intestinal epithelial cells (IECs) and intestinal organoids from mice with an enterocyte specific *Zip14* (Slc39a14) deletion.

## 2. Materials and Methods

### 2.1. Animals 

Whole Body Zip14−/− (WB-KO) mice were developed, characterized, and backcrossed as reported previously [18]. Wild-type (WT) mice of the same strain were used as controls. Heterozygous mutant (−/+) × heterozygous mutant (−/+) mating scheme was used to generate both genotypes for experiments. Enterocyte-specific *Zip14* knockout mice (*Zip14*ΔIEC) were generated by targeted deletion of introns 4–8 [19]. Briefly, founder *Zip14*^flox/+^ mice were bred to obtain a floxed *Zip14* (*Zip14*^F/F^) mouse line. They were crossed with Villin-Cre (VC-*Zip14*+/+) mice of the C57BL/6 line from The Jackson Laboratory (JAX # 004586; Bar Harbor, ME, USA) to create enterocyte-specific *Zip14* conditional knockout (*Zip14*^ΔIEC^) mice. *Zip14*^F/F^ mice were used as controls for experiments with this strain. Breeding and genotyping were conducted at the University of Florida (Gainesville, FL, USA). Starting at weaning, the mice were fed ad libitum a commercial diet (Teklad LM-485 Mouse/Rat Sterilizable Diet; Harlan 7012, ENVIGO, Madison, WI, USA) and tap water. A 12 h light/dark cycle was used, and the mice were maintained in plastic barrier caging. Young adult male and female mice (8–16 weeks of age) were used for the experiments. Mice were killed by exsanguination, using cardiac puncture under isoflurane anesthesia, between 11:00 am and 2:00 pm. Protocols were approved by the University of Florida Institutional Animal Care and Use Committee (protocol # 202007015).

### 2.2. Hematoxylin and Eosin Staining (H&E) 

Mice intestinal tissues were fixed in 4% paraformaldehyde for 24 h, embedded in paraffin, cut into 5-μm thick sections, dewaxed in xylene, and rehydrated with ethanol. Sections were then stained with hematoxylin for 2 min, followed by eosin for 2 min. Light microscopy was used for histological and pathological analysis at the UF molecular pathology core. Histopathological scores of intestine staining were assigned in a blinded manner.

### 2.3. Tissue Preparation and RNA Isolation 

Tissues (liver, lung, pancreas, colon, gastrocnemius muscle, intestine, spleen, kidney) from enterocyte-specific and WB-KO mice were excised and samples were flash-frozen in liquid nitrogen (N_2_) and stored at −80 °C prior to RNA extraction. Homogenization with TRI Reagent® (Molecular Research Center, Cincinnati, OH, USA) was performed with a Bullet Blender® (Next Advance, Raymertown, NY, USA). IECs were isolated from the first 20 cm of the intestine, after perfusion with ice-cold buffer (10 mM EDTA, 10 mM HEPES, 0.9% NaCl), using a chelation method [20]. For RNA-seq analysis, RNA from sections of proximal small intestine of WB-KO and WT mice was extracted as above. All RNA was isolated using RNeasy^®^ reagents and spin columns (Qiagen #217604, Hilden, Germany). RNA samples were treated with deoxyribonuclease (DNase). Quantity and quality of the RNA were assessed spectrophotometrically and further analyzed for integrity using a Bioanalyzer (Agilent, Santa Clara, CA, USA). Only RNA with an integrity number of at least 6.8 were used for RNA-sequencing (RNA-seq). 

### 2.4. RNA-Sequencing 

RNA-seq libraries, prepared from samples from individual mice, were sequenced on an Illumina Nova Seq 6000 Platform at Novogene Corp. (Sacramento, CA, USA). Genes with a Log2 fold change (FC) of >2, and lncRNAs and miRNAs with an FC of >1 were used for differential expression analysis [21,22]. RNA Sequencing data files were deposited into Gene Expression Omnibus—NCBI (accession number GSE210160).

### 2.5. Quantitative PCR 

For qPCR first-strand cDNA synthesis, 2 µg of RNA was reversed-transcribed using High-Capacity RNA-to-DNA™ reagents (Applied Biosystems, Waltham, MA, USA). EXPRESS SYBR™ GreenER™ Supermix with ROX, or specific TaqMan assays (ThermoFisher, Waltham, MA, USA), were used for qPCR reactions; all following the manufacturer’s instructions. Primer/probe sequences were reported previously [17] or are reported in the Appendix A. *H19*, *miR-675-3p* and *miR-675-5p* qPCR assays used primer/probe sets from GeneCopoeia (Rockville, MD, USA). A QuantStudio 3 instrument (ThermoFisher, Waltham, MA, USA) was used for sequence detection. qPCR assays were run in duplicate, with relative expression calculated by the 2^−ΔΔCt^ method. All assays were normalized to *18S* rRNA, except for *miR-675* RNAs, which used *U6* RNA.

### 2.6. Chromatin Immunoprecipitation (ChIP) Assays

ChIP assays were performed using intestinal epithelial cells from enterocyte-specific *Zip14* knockout mice [19]. Briefly, tissue or cells were placed in PBS containing 1% formaldehyde for crosslinking. The crosslinking reaction was stopped by incubating in 20X glycine (0.2 M Tris, 1.5 M glycine (pH 8.0)) for 5 min. After washing, the chromatin was re-suspended in sonication cell lysis buffer and fragmented by sonication (Bioruptor^®^ sonicator (Diagenode, Denville, NJ, USA)). The chromatin nuclear extract was immunoprecipitated following the manufacturer’s protocol (Cell Signaling Technology, Danvers, MA, USA). The antibodies used were nuclear factor kappa beta (NF-ĸβ) (#8242) and signal transducer and activator of transcription 3 (STAT3) (#4904) from Cell Signaling Technology (Danvers, MA, USA). The primers used for *H19*, *Meg3* and *U90926* promoters are described in the Appendix A.

### 2.7. Preparation and Treatment of Intestinal Organoids (Enteroids)

A 20 cm segment of proximal small intestine from both *Zip14*^ΔIEC^ and *Zip14*^F/F^ mice were excised and perfused with phosphate-buffered saline (PBS), opened longitudinally, cut into 5 mm lengths and placed in Cell Dissociation Reagent (Stemcell Technologies, Vancouver, BC, Canada). The domes produced, were seeded into Matrigel^®^ (Corning, Corning, NY, USA) and incubated for 20 min at 37 °C, with InstestiCult™ (Stemcell Technologies, Vancouver, Canada) added thereafter. The enteroids were cultured for 10 days. These media were replaced every three days. Enteroids in the Matrigel domes were placed in Organoid Harvesting Solution (Cultrex, R&D Systems, Inc., Minneapolis, MN, USA), washed with PBS, and RNA was isolated using RNeasy^®^ and TRI Reagent^®^ as above.

### 2.8. Statistics

Data are presented as means ± standard error of the mean (SEM) of biological replicates. Prism 6.0 software (Graph Pad Software Inc., La Jolla, CA, USA) was used for these analyses. Differences between genotypes were evaluated using Student’s *t*-test and multiple comparisons used ANOVA, followed by Tukey’s test. Statistical significance was indicated as * *p* < 0.05; ** *p* < 0.01; *** *p* < 0.001; **** *p* < 0.0001; ns = not significant.

## 3. Results

### 3.1. Influence of WB-Zip14 Ablation on Differential Expression of Specific lncRNA Genes in Proximal Small Intestine

Transcriptome profiling was carried out to examine the influence of *Zip14* ablation in small intestine. A WB-*Zip14* knockout model was used, with profiling by DNA array and RNA-sequencing, respectively [17,23]. Focusing on the small intestine of mice, RNA-seq was used to profile changes in lncRNA and miRNA transcripts. The volcano plot (Figure 1A) shows that many lncRNAs had expression differentials (Fold Change) of >1.0 or <−1.0. A total of 53 and 55 lncRNAs met those criteria, respectively. Only five miRNAs had a FC of >1.0 and none were <−1.0. The full list, including cDNA sequences, predicted genes and uncharacterized cDNAs is presented as Appendix A. Read densities for *H19*, *U90926* and *Meg3* (Table 1) display the marked differences in lncRNA sequences produced by WB-*Zip14* ablation (Figure 1B). Note read densities for *Malat1*, used as a control, are uniform compared to those of the differentially expressed lncRNAs. A number of transcripts identified as upregulated were selected and confirmed as upregulated (Figure 1C). To assess whether *Zip14* ablation is associated with a specific mucosal pattern of inflammation, H&E-stained histological sections from intestinal tissue (duodenum) were examined. Histological examination of the small intestine showed no difference in the number of crypts, villus and goblet cell between both genotypes (Appendix A).

### 3.2. Tissue Specific Expression of lncRNAs in Mice with WB-Zip14 Deletion

To compare tissue-specificity of lncRNA and miRNA expression changes produced by *Zip14* ablation, we used RNA from liver, lung, pancreas, colon, muscle, intestine, isolated IECs and enteroids from mice of the WB-KO and WT genotypes. *U90926, H19* and *Meg3* were selected as representing a range of FC differences. *U90926* was differentially expressed in intestinal tissue, colon and IECs, but was undetectable in the other tissues examined, except WT lung (Figure 2A). *H19* mRNA was detected in liver, muscle, enteroids, tissues of the gastrointestinal (GI) tract, and WT lung (Figure 2B). Similarly, *Meg3* is highly upregulated in the GI tract (Figure 2C). Of note is that neither *Meg3* nor *U90926* are detectable with enteroid cultures (Figure 2A, C). Overall, these lncRNAs were found to be the most differentially expressed in the GI tract compared to the other tissues in our WB-KO mice.

### 3.3. Tissue Specific Expression of miR-7027

Low expression of *miR-7027* was detected in muscle, liver, kidney, spleen and lung regardless of WB-*Zip14* ablation (Figure 3A). However, increased expression of this miRNA was robustly upregulated in intestine and IECs of mice with WB-*Zip14* deletion. Furthermore, both *miR-7027-5p* and *miR-7027-3p* levels are significantly (*p* < 0.05/0.01) increased in *Zip14*^ΔIEC^ mice (Figure 3B). 

### 3.4. Tissue Specific Expression of lncRNAs in Mice with Zip14^ΔIEC^ Deletion

qPCR assays of proximal small intestine and muscle RNA from *Zip14*
^ΔIEC^ (ΔIEC) and *Zip14*^F/F^ (F/F) control mice were compared to visualize the changes in lncRNA expression. *H19*, *Meg3*, *U90926*, *Bvht*, and *Pvt1* levels were significantly (*p* < 0.05, 0.01) increased in intestine of ΔIEC mice (Figure 4A). In contrast, none of these lncRNAs were affected in skeletal muscle with this enterocyte specific *Zip14* deletion (Figure 4B). Normalization of 1.0 indicates a detectable signal rather than non-detectable qPCR signals.

### 3.5. Promoter Occupancy May Explain Differential Expression of H19, Meg3 and U90926 with Zip14 Ablation

Bioinformatic searches revealed promoters of *Meg3*, *H19* and *U90926* all have consensus binding sites for the transcription factors STAT3 and NF-ĸβ. ChIP assays using chromatin derived from IECs, isolated from ΔIEC and F/F mice, were conducted. No difference in binding of STAT3 were noted for the *H19* promoter (Figure 5A). Increased binding of this TF in one of the identified STAT3 sites in both the *Meg3* and *U90926* promoters was detected. There was increased binding of NF-ĸβ at two sites of the *H19* promoter and one site of the *Meg3* promoter (Figure 5B). Decreased NF-ĸβ binding was detected on the *U90926* promoter in the ΔIEC mice (Figure 5B). These differences suggest that the upregulation of these genes in the ΔIEC mice could be a proinflammatory response produced by these TFs through loss of *Zip14*.

## 4. Discussion

Ablation of *Zip14* in mice leads to diminished transport of zinc into enterocytes [17,24], and produces interesting phenotypes including decreased intestinal barrier function, adipose hyperplasia, muscle wasting, glucose processing defects, and skeletal defects with aging [25]. Mutations of *Zip14* in humans leads to neurological defects and skeletal abnormalities [15,16]. 

Our approach to examine a biochemical basis of Zip14 function has been through transcriptome comparisons of mice with deletions of the Zip14 gene. Whole body deletion of *Zip14* yields upregulation and down-regulation of specific genes in muscle [17]. Surprisingly, among the most upregulated genes were *H19*, and *miR-675*, which is derived from *H19*. These findings confirmed those of others; *H19* is not silenced at birth in muscle and is expressed in mice into adulthood [16,26]. We also showed that LPS administration induced *miR-675* in muscle, a response that was markedly enhanced in whole body *Zip14* KO mice [17]. LPS administration also induces Zip14 and increases zinc uptake into muscle of WT mice, but not in WB-KO mice. Since *Zip14* ablation leads to enhanced binding of inflammation-related TFs in muscle, we concluded the elevated *miR-675* is a reflection of loss of a mechanism associated with muscle wasting [17]. We propose that the ablation of Zip14 function leads to upregulated expression of lncRNAs and miRNAs that in turn regulate specific inflammation-related genes.

The influence of nutrition on epigenetic changes and its relationship with lncRNAs or miRNAs modification has not been widely studied [27]. Minerals such as zinc are known to have the ability to modify epigenetic markers and alter cell signaling during inflammation or intestinal homeostasis [28,29]. The in-silico integrative analysis of the lncRNAs and miRNAs (RNA-seq analysis data) shows that the possible target genes of these miRNAs are related to the signaling pathways of transcriptional factors and inflammation pathway mediated by cytokines and chemokines. Zinc, as a signaling molecule, is critical to maintain proliferative cellular capacity that allows self-renewal of intestinal cells to maintain homeostasis of the intestinal epithelium [30]. The cytokine- and chemokine-mediated inflammation pathway includes key protein/lncRNAs/miRNAs in intercellular communication that help maintain intestinal homeostasis [31], as well as driving intestinal inflammation in various pathological processes [32,33].

There have been relatively few studies on zinc/zinc transporters and lncRNAs/miRNAs. These have focused on miRNAs and genes controlling zinc metabolism [16], miRNAs/ human dietary zinc restriction [10], zinc deficiency/cancer [34] and specific zinc transporters, Zip8 [35], Zip7 [13], and ZnT4 [14].

We have focused in this experimental series on the small intestine, as *Zip14* is most highly expressed in the proximal region and decreases along the gastrointestinal axis [24]. Furthermore, *Zip14* deletion in the intestine leads to decreased barrier function and mild endotoxemia [36]. There is considerable interest in lncRNAs in intestinal physiology [37]. Intestinal miRNAs are important factors in the function and regeneration of the intestinal epithelium. They have an important role in gene regulation under pathological and physiological conditions, including inflammatory response. RNA-seq differential expression analysis showed three highly expressed miRNAs including *miR-99agh* and *-7027* (Appendix A). A role for *miR-7027* in the regulation of intestinal inflammation and gene regulation has not previously been demonstrated. After in-silico analysis, a relevant function has been proposed for *miR-7027* in the intestinal mouse epithelium where this miRNA directly regulates inflammation by binding to 3′ untranslated region (3’UTR) of STAT3 and interleukin-21 (IL-21) genes. Upregulation of these genes has been shown by RNA-seq analysis (GSE210160). *miR-7027* is differentially expressed in both crypts and villus and was reported to be associated with pathological disorders [38]. It also has a compensatory function under the ablation or downregulation of other miRNAs [39].

As has been shown numerous times, and again here, *H19* is expressed in the intestine. Functions related to intestinal permeability and inflammation have been shown where this lncRNA acts to regulate specific mRNAs [40,41]. *Meg3* has been implicated in LPS- induced intestinal injury [42] and is believed to be upregulated by STAT3 in cardiomyocytes [43]. This latter finding agrees with results reported here. Expression of *U90926* in intestine has been reported [44] and elevated intestinal expression has been demonstrated as a response to Giardia infection [45]. It has been proposed that lncRNAs in intestine are a reflection of resident consensual microbiota [46]. As we have found, *Zip14* deletion increases the abundance of microbes of the Akkermansia phylum (In-press). Hence deletion of a metal transporter that supplies zinc to enterocytes may alter intestinal microflora that provide signals regulating lncRNA production. In addition, the bulk of the literature on enteric lncRNAs relate to cancer, thus the loss of zinc transport through *Zip14* deletion could establish precancerous conditions. 

## 5. Conclusions

The consistent and precise finding of the genomic driver of intestinal homeostasis and remodeling may have a critical implication for intestinal diseases and nutrition. The experiments reported here demonstrate deletion of the metal transporter Zip14 gene initiates the upregulation of specific lncRNAs and miRNAs, as detected by both RNA-Seq and specific qPCR assays. Enterocyte-specific ablation of *Zip14* restricts changes in those RNAs to the intestine. The binding of proinflammatory TFs, NF-ĸβ and STAT3, to the *H19*, *Meg3*, and *U90926* promoters is consistent with earlier work where Zip14-induced epigenetic factors influence TF occupancy of specific regulatory sites. These findings provide supporting evidence on the role of the Zip14 transporter, as its ablation initiates an inflammatory cascade of responses by activating noncoding regulatory elements in the intestinal epithelium. The *Zip14* transcriptome study of comparative patterns in the intestinal epithelium is valuable to better understand the function and regulation of human orthologous genes during intestinal diseases. Nevertheless, future studies are needed to perform further validation of the functions of lncRNAs and microRNAs in response to intestinal microbiota, nutrition, and disease.

## Figures and Tables

**Figure 1 nutrients-14-05114-f001:**
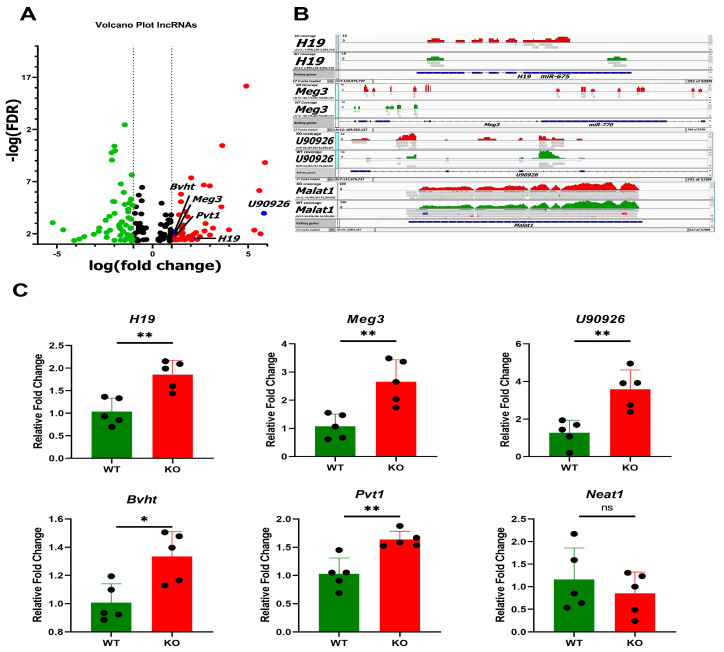
Effect of WB-*Zip14* knockout on differential expression of lncRNAs in proximal small intestine. (**A**) Volcano plot of RNA-sequencing showing transcription changes with *Zip14* ablation. (**B**) Read densities of *U90926*, *H19*, *Meg3* and *Malat1* from RNA-Seq. (**C**) qPCR assay validation of differential expression in *H19*, *Meg3*, *U90926*, *Bvht* and *Pvt1*. *Neat1* was not confirmed as differentially expressed. Values are reported as the mean ± SEM. *n* = 5 (each dot indicates data measurement for one mouse). * *p* < 0.05, ** *p* < 0.01, ns = non-significant. Student’s *t* test for WT and Zip14 KO comparison. qPCR = quantitative polymerase chain reaction; KO = knockout mice; WT = wild-type mice.

**Figure 2 nutrients-14-05114-f002:**
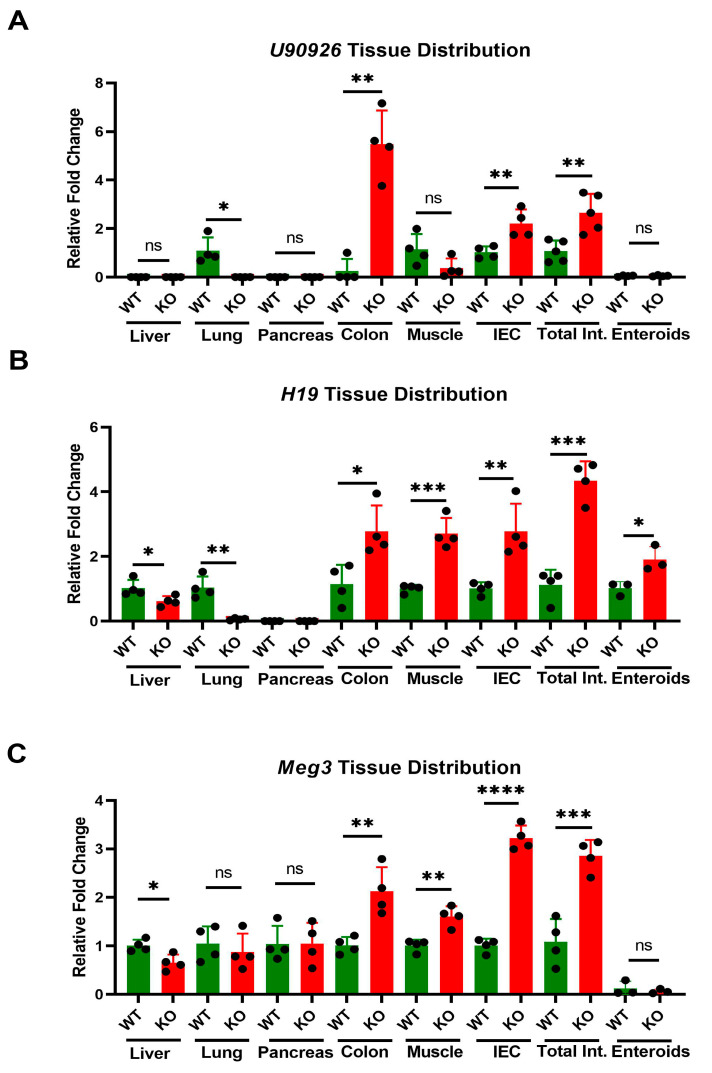
Effect of WB-*Zip14* knockout on lncRNAs in selected tissues as measured by qPCR. (**A**) *H19*; (**B**) *Meg3*; (**C**) *U90926*. Values are reported as the mean ± SEM. *n* = 3–4 (each dot indicates data measurement for one mouse). * *p* < 0.05; ** *p* < 0.01; *** *p* < 0.001; **** *p* < 0.0001; ns = non-significant. Student’s *t* test for WT and Zip14 KO comparison. Int = Intestine; IEC = intestinal epithelial cells.

**Figure 3 nutrients-14-05114-f003:**
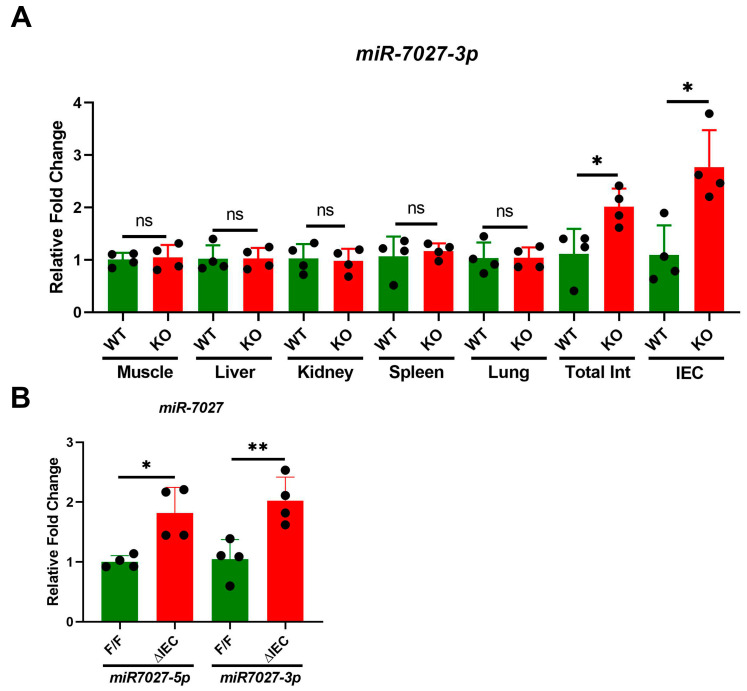
Effect of *Zip14* knockout on *miR-7027* as measured by qPCR. (**A**) *miR-7027-3p* in selected tissues from mice with WB *Zip14* knockout. (**B**) Level of *miR-7027-3p* and *miR-7027-5p* in enterocytes from mice with enterocyte-specific *Zip14* knockout. Values are reported as the mean ± SEM. *n* = 4 (each dot indicates data measurement for one mouse). * *p* < 0.05; ** *p* < 0.01; ns = non-significant. Student’s *t* test for WT vs. *Zip14* KO and F/F, vs. ∆IEC comparisons. F/F = floxed/floxed mouse; ∆IEC = Enterocyte-specific *Zip14* knockout mice; Int = Intestine; IEC = intestinal epithelial cells.

**Figure 4 nutrients-14-05114-f004:**
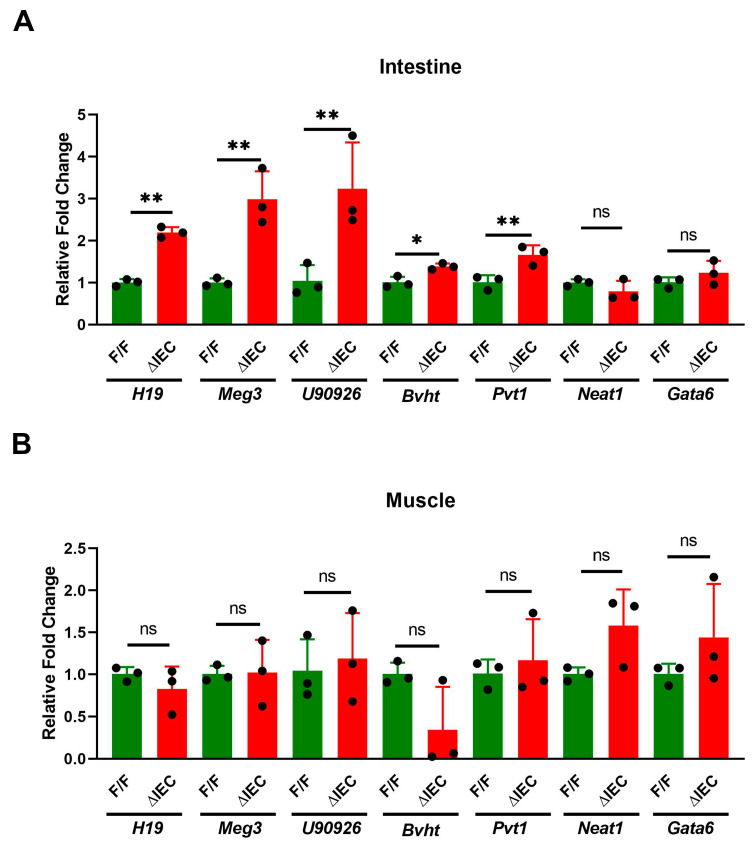
Effect of enterocyte specific *Zip14* knockout on lncRNAs. (**A**) Proximal intestine; (**B**) Skeletal muscle. Values are reported as the mean ± SEM. *n* = 3 (each dot indicates data measurement for one mouse). * *p* < 0.05; ** *p* < 0.01; ns = non-significant. Student’s *t* test for F/F and ∆IEC comparison. F/F = floxed/floxed mouse; ∆IEC = Enterocyte-specific *Zip14* knockout mice.

**Figure 5 nutrients-14-05114-f005:**
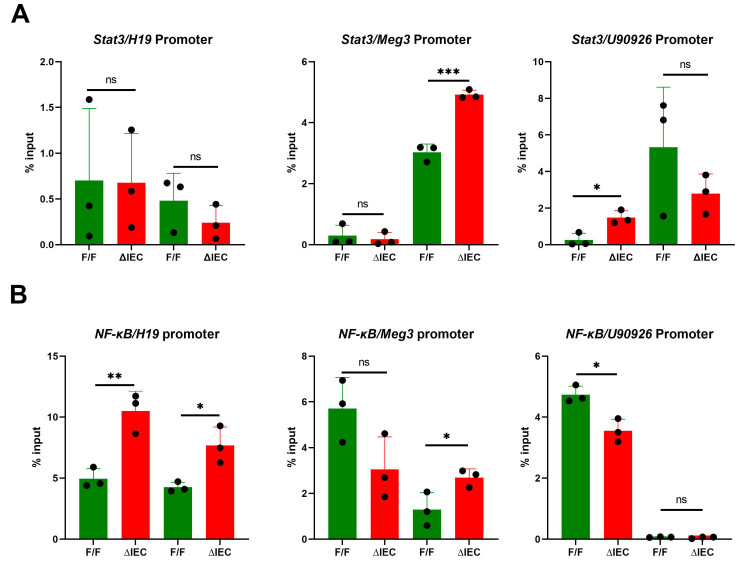
ChIP assays of STAT3 (**A**) and NF-κβ (**B**) binding to promoters for *H19*, *Meg3* and *U90926* using chromatin from enterocyte-specific *Zip14* knockout mice. Values are reported as the mean ± SEM. *n* = 3 (each dot indicates data measurement for one mouse). * *p* < 0.05; ** *p* < 0.01; *** *p* < 0.001; ns = non-significant. Data were analyzed using Student’s *t* test for F/F and ∆IEC comparison. F/F = floxed/floxed mouse; ∆IEC = Enterocyte-specific *Zip14* knockout mice; ChIP = chromatin immunoprecipitation; STAT3 = signal transducer and activator of transcription 3; NF-ĸβ = nuclear factor kappa beta.

**Table 1 nutrients-14-05114-t001:** Upregulated lncRNAs. RNA-seq was used to profile changes in lncRNA expression.

Symbol	log2FoldChange	LOG(FDR)	padj	*p*-Value	Name
*U90926*	5.826542	3.95	0.000111	8.0 × 10^6^	cDNA sequence U90926
*H19*	2.107413	1.51	0.030793	7.5 × 10^3^	H19, imprinted maternally expressed transcript
*Bvht*	1.132362	2.15	0.007004	1.2 × 10^3^	Braveheart long non-coding RNA
*Pvt1*	1.052021	1.66	0.021957	4.8 × 10^3^	Plasmacytoma_variant_translocation_1
*Meg3*	1.044219	1.60	0.025132	5.8 × 10^3^	Maternally expressed 3
*Neat1*	−0.009449	0.68	0.968169	9.9 × 10^1^	Nuclear paraspeckle assembly transcript 1 (non-protein coding)

padj = adjusted *p*-value; FDR = False discovery rate.

## Data Availability

All data are available in the text of Appendix A.

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
