# Peer review of "Long Noncoding RNA, MicroRNA, Zn Transporter Zip14 (Slc39a14) and Inflammation in Mice"

_nutrients, 2022, doi:10.3390/nu14235114_

Round 1

Reviewer 1 Report

The article entitled Long Noncoding RNA, MicroRNA, Zn Transporter Zip14 2 (Slc39a14) and Inflammation in Mice, has taken into consideration the interesting medical problem e.i.: the role of zinc transporter Zip14. As the authors right noted the lack of Zip14 can lead to proinflammatory states. Moreover, the zinc effect has been investigated in the context of several cancers type and neurodegenerative disorders. For their studies presented in the manuscript authors used several adequate techniques including the special mice types, RNA-sequencing, quantitative PCR, etc. However, I did not find any information on how big was mouse investigated and the control group as well as the identity of the bioethical commission agreement. The above is strongly required for these studies.

The second question is the connection of this manuscript with nutrition - I think that authors should put effort to justify their studies in this field. Moreover, the importance or medical application of presented results for human health and safety nutrition is strongly demanded.

In conclusion at present form I cannot recommend the article for publication but I expect that the authors provide some suitable corrections for the further consideration

Author Response

Reviewer 1:

The article entitled Long Noncoding RNA, MicroRNA, Zn Transporter Zip14 2 (Slc39a14) and Inflammation in Mice, has taken into consideration the interesting medical problem e.i.: the role of zinc transporter Zip14. As the authors right noted the lack of Zip14 can lead to proinflammatory states. Moreover, the zinc effect has been investigated in the context of several cancers type and neurodegenerative disorders. For their studies presented in the manuscript authors used several adequate techniques including the special mice types, RNA-sequencing, quantitative PCR, etc. However, I did not find any information on how big was mouse investigated and the control group as well as the identity of the bioethical commission agreement. The above is strongly required for these studies.

Previously we stated the mice are 8-16 weeks old (Line 82-83). This puts their weight at about 25 grams. The mice of the genotypes used are comparable and we age match them for experiments. We use a relatively simple mating scheme of het x het to produce our knockout and control mice from the same strain. This information has been added (Line 73-74). The bioethical concerns are not relevant since the study design was approved by our IACUC. The approval number was added to the manuscript (Line 86).

The second question is the connection of this manuscript with nutrition - I think that authors should put effort to justify their studies in this field. Moreover, the importance or medical application of presented results for human health and safety nutrition is strongly demanded.

The relevance to nutrition is clear. Zinc is a micronutrient and needs to be transported to be functional (Lines 37 to 42). More information pertaining to nutrition has been added to the manuscript (Lines 339-350). It should be mentioned that this paper was requested by an editorial member for a specific supplement for Nutrients.

In conclusion at present form I cannot recommend the article for publication but I expect that the authors provide some suitable corrections for the further consideration

Thank you for your input. We made the corrections mentioned for further consideration. 

Reviewer 1 Report Form:

More background information and references have been added to the Introduction.

We believe the references are appropriate and modest in number.

It is difficult to add anything new at this point in lncRNA research.

We have added more descriptions in the Materials and Methods section. We were trying to be brief.

We have added more information to clarify the Results.

We added a little to the Conclusion and believe it is modest and appropriate at this stage.

Reviewer 2 Report

1. Results 1 Transcriptome profiling was carried out to examine the influence of Zip14 ablation in small intestine. However, pathological analysis needs to be supplemented to confirm whether Zip14 ablation will affect the normal structure of the small intestine.

2. Why these tissues (liver, lung, pancreas, colon, muscle, isolated IECs, total intestine isolated and enteroids (intestinal organoids) ) were selected for further analyzed instead of stomach, heart, etc?

3.  What is the different of isolated IECs, total intestine isolated and enteroids (intestinal organoids)? Why is intestinal tissue classified in such detail?

4. Why miR-7027 was selected to further analyzed in different tissues, is miR-7027 belonged to the differentially expressed results in figure 1?

5. Only the differential analysis of expression profile was conducted, and the analysis of disease occurrence or pathological changes was not conducted after Zip14 ablation, so the significance of the manuscript was not great.

Author Response

Reviewer 2:

  1. Results 1 Transcriptome profiling was carried out to examine the influence of Zip14 ablation in small intestine. However, pathological analysis needs to be supplemented to confirm whether Zip14 ablation will affect the normal structure of the small intestine.

We added a histological description and photomicrographs in the supplementary materials to confirm that Zip14 ablation does not affect the intestinal mouse morphology or pathology (Lines 171-174; Figure S1).

  1. Why these tissues (liver, lung, pancreas, colon, muscle, isolated IECs, total intestine isolated and enteroids (intestinal organoids) ) were selected for further analyzed instead of stomach, heart, etc?

Time permitted to harvest the tissues was limited to minimize degradation, so we chose those mentioned. The GI tract was our main focus and the other tissues selected express Zip14 and show changes with Zip14 deletion.

  1. What is the different of isolated IECs, total intestine isolated and enteroids (intestinal organoids)? Why is intestinal tissue classified in such detail?

Each preparative method provides a different purity for enterocytes. Total intestine is an excised section of the small intestine. IECs are isolated epithelial cells from the intestine tissue. Enteroids were collected from the intestine crypts and plated in a matrix to form an organ-like cell. Enteroids allow treatments to be introduced during an experiment. For example, we treated enteroids with 15µM zinc sulphate for 16hrs, but the lncRNA/ miRNA inflammatory response did not change (in press). These three forms of intestine were classified and compared in detail to emphasize that the IEC is the site of these changes in lncRNAs expression.

  1. Why miR-7027 was selected to further analyzed in different tissues, is miR-7027 belonged to the differentially expressed results in figure 1?

We included miR-7027 because it has been characterized by others and it was highly upregulated in our RNA-Seq profile. We added more information about miR-7027 to the manuscript to clarify our interest in this miRNA (Lines 361-370). miR-7027 was not included in Figure 1 because it is a microRNA, whereas the targets of Figure 1 are long non-coding RNAs (lncRNAs).

  1. Only the differential analysis of expression profile was conducted, and the analysis of disease occurrence or pathological changes was not conducted after Zip14 ablation, so the significance of the manuscript was not great.

Mutations have been reported for human Zip14 that lead to sever pathology (Hendrickx, 2018 doi: 10.1371/journal.pgen.1007321; Wang G, 2018 doi: 10.1038/s41591-018-0054-2). Most likely other functional defects will emerge as the field expands.

Reviewer 2 Report Form:

The manuscript has been spell checked.

More information has been included in the Introduction, Methods, and Discussion. We were attempting to be concise.

Round 2

Reviewer 1 Report

The Authors have provided sufficient corrections which make the article suitable for publication.

Reviewer 2 Report

I think this manuscript was well organized and it could be accepted.